# Confining H$_3$PO$_4$ network in covalent organic frameworks enables proton super flow

Shanshan Tao[1], Lipeng Zhai[1], A. D. Dinga Wonanke[2], Matthew A. Addicoat [2], Qiuhong Jiang[1] & Donglin Jiang[1,3]✉

Development of porous materials combining stability and high performance has remained a challenge. This is particularly true for proton-transporting materials essential for applications in sensing, catalysis and energy conversion and storage. Here we report the topology guided synthesis of an imine-bonded (C=N) dually stable covalent organic framework to construct dense yet aligned one-dimensional nanochannels, in which the linkers induce hyperconjugation and inductive effects to stabilize the pore structure and the nitrogen sites on pore walls confine and stabilize the H$_3$PO$_4$ network in the channels via hydrogen-bonding interactions. The resulting materials enable proton super flow to enhance rates by 2–8 orders of magnitude compared to other analogues. Temperature profile and molecular dynamics reveal proton hopping at low activation and reorganization energies with greatly enhanced mobility.

[1] Department of Chemistry, Faculty of Science, National University of Singapore, 3 Science Drive 3, Singapore 117543, Singapore. [2] School of Science and Technology, Nottingham Trent University, Clifton Lane, Nottingham NG11 8NS, UK. [3] Joint School of National University of Singapore and Tianjin University, International Campus of Tianjin University, Binhai New City, Fuzhou 350207, China. ✉email: chmjd@nus.edu.sg

One-dimensional (1D) single pore, as observed for proton channels when the pore width is small enough (<3.5 Å, submicropore) to confine a single file proton chain, can trigger proton super flow[1,2]. Such a high-rate transport is not only important for maintaining biological functions but also ideal for sensing, catalysis and energy conversion and storage. However, for applications, a material must consist of dense and large 1D channels other than a single submicropore. In this context, porous frameworks are promising as they are capable of developing built-in channels[3–6]. In synthetic systems, proton conduction can be categorised into two different classes; one is proton conduction under humidity using water as proton carrier and another is anhydrous proton conduction free of water. Porous materials have been extensively explored for water-mediated proton conduction as hydrogen-bonding network in water facilitates proton transport, which however can work only at temperatures below 100 °C[5–10]. In contrast, anhydrous proton conduction is typically based on organic heterocyclic compounds or pure phosphoric acid that are representative of proton carriers other than water, which enable high temperature proton transport but require extraordinary stability of the porous materials and usually finish with a much lower rate. Thus, designing a material that can combine high stability and anhydrous proton conductivity is a challenging goal.

We sought to explore an anhydrous proton-conducting system as we anticipate synthetic channels with large pore size could become a freeway for proton super flow. We focus on porous molecular frameworks, specifically covalent organic frameworks (COFs)[11–16] as predesignable yet stable materials for proton transport. We reasoned that COFs could potentially achieve both stability and proton conductivity: (i) the framework stability can be assumed using strong covalent bond as linkage, while designing building units with a capability of triggering inter and intralayer interactions can further improve stability;[17,18] (ii) the stability of proton network loaded in channels can be realised by anchoring the network on pore walls, which can be preset with specific sites to enable interactions with the proton network;[6] (iii) the frameworks can be constructed via topology-guided growth of polygonal backbones to achieve aligned yet dense 1D channels together with a simultaneous manipulation of pore shape, size and connectivity;[11,12,15] (iv) using pure organic units with lightweight elements such as H, C and N offers the possibility to synthesise highly porous material that enables exceptional loading of proton network in the channels[11,13]. We unexpectedly found that such well-designed large 1D channels offer the structural base for confining proton network to enable super flow across the material.

## Results

**Design and synthesis of stable COFs.** We designed and synthesised a dually stable crystalline porous TPB-DMeTP-COF (Fig. 1a) by a topology-guided polycondensation of $C_3$-symmetric 1,3,5-tri(4-aminophenyl)benzene (TPB) as knot and $C_2$-symmetric 2,5-dimethylterephthalaldehyde (DMeTP) as linker under solvothermal conditions to constitute hexagonal 1D open channels (o-DCB/n-BuOH (0.5 mL/0.5 mL), acetic acid (0.1 mL, 6 M), 120 °C and 3 days). We screened the polycondensation conditions to achieve the highest crystallinity and porosity by tuning monomer concentration, solvent, acid and temperature; the COF porosity predetermines the density of the 1D nanochannels as well as the content of proton carrier in the material. The methyl groups on the phenyl linkers (Fig. 1a, inset) trigger hyperconjugation and inductive effects that weaken the polarisation of C=N units and soften the interlayer charge repulsion, yielding an exceptionally stable framework[13]. TPB-DMeTP-COF uses

lightweight elements of H, C and N to form dense channels with a pore width of 3.36 nm (Fig. 1a), while the hexagonal scaffold is shape persistent and covalently linked by strong C=N bonds (bonding energy = 6.38 eV; Fig. 1a).

As one macrocycle consists of six inward $N_{in}$ sites from the C=N units (Fig. 1b), stacking along the $z$ direction at an interval of 3.52 Å constitutes six $N_{in}$ chains on the six pore walls (Fig. 1c, only 20 layers are shown) that can anchor proton network in the pore. This multichain multipoint anchoring effect stabilises the proton network in the channels. Moreover, the macrocycle is further linked through the $C_3$-symmetric knots to form extended two-dimensional (2D) hexagonal sheet that stacks to create aligned yet dense channels across the material (Fig. 1d). A distinct feature is that the 1D channels extend straight and are independent from each other (Fig. 1d), which are distinct from more conventionally obtained amorphous analogue with chain entanglement and pore interpenetration.

We selected neat $H_3PO_4$ as proton carrier for loading into the nanochannels, as it is nonvolatile and nontoxic and forms hydrogen-bonding network via P=O···H–O and P–O···H–O interactions[19]. Moreover, $H_3PO_4$ has the highest proton conductivity among all the materials[19,20], which underpin broad interest in developing $H_3PO_4$-based proton-transporting systems[3,5,6,21]. Nevertheless, a porous material in conjunction with $H_3PO_4$ that can combine stability and high performance is still inaccessible. In contrast, TPB-DMeTP-COF is distinct, as the six $N_{in}$ chains can anchor and reinforce the $H_3PO_4$ network in the channels; the overall $N_{in}$···H–O hydrogen-bonding force between the $N_{in}$ chains and the $H_3PO_4$ network are extremely strong (vide infra), as the interactions occur across the whole channel in a three-dimensional multichain multipoint mode. Therefore, TPB-DMeTP-COF stabilises not only its own pore structure but also the $H_3PO_4$ networks in the channels, creating a dually stable material.

**Crystal structure and characterisation.** The TPB-DMeTP-COF exhibited the powder X-ray diffraction (PXRD) peaks at 2.82°, 4.94°, 5.62°, 7.54°, 10.1° and 25.4°, which were assigned to the (100), (110), (200), (210), (220) and (001) facets, respectively (Fig. 2a, red curve and inset). We used density-functional based tight-binding (DFTB+) calculations to optimise the conformation of single 2D sheet and the configuration of stacking modes[22]. The AA-stacking mode (Fig. 2a, blue curve) reproduces the PXRD peak position and intensity. The TPB-DMeTP-COF assumes a space group of $P_6$ with unit cell parameters of $a = b = 37.2718$ Å, $c = 3.52$ Å, $\alpha = \beta = 90°$ and $\gamma = 120°$ (atomistic coordinates see Supplementary Table 1). We conducted Pawley refinements (Fig. 2a, green curve) and confirmed the correctness of the PXRD peak assignment, as indicated by a negligible difference (Fig. 2a, black curve) with $R_p$ and $R_{wp}$ values of 1.12% and 2.01%, respectively (Supplementary Table 2). Rietveld refinement also yields a $P_6$ space group (Supplementary Table 3). Therefore, TPB-DMeTP-COF consists of an extended hexagonal lattice with dense mesoporous 1D channels with pore width of 3.36 nm (Fig. 2b). The presence of the (001) facet at 25.4° indicates the structural ordering with 3.52-Å separation along the $z$ direction (Fig. 2c). This reflects that the $N_{in}$ sites are highly dense on the channel walls as they are separated by only 3.52 Å. The TPB-DMeTP-COF exhibited a reversible nitrogen sorption isotherm (Fig. 2d) and is highly porous to achieve a Brunauer–Emmett–Teller (BET) surface area of 2894 $m^2\,g^{-1}$, a Langmuir surface area of 4586 $m^2\,g^{-1}$ (Supplementary Table 4), a pore size of 3.36 nm (Fig. 2e, circles) and an exceptional pore volume of 1.60 $cm^3\,g^{-1}$ (Fig. 2e, squares).

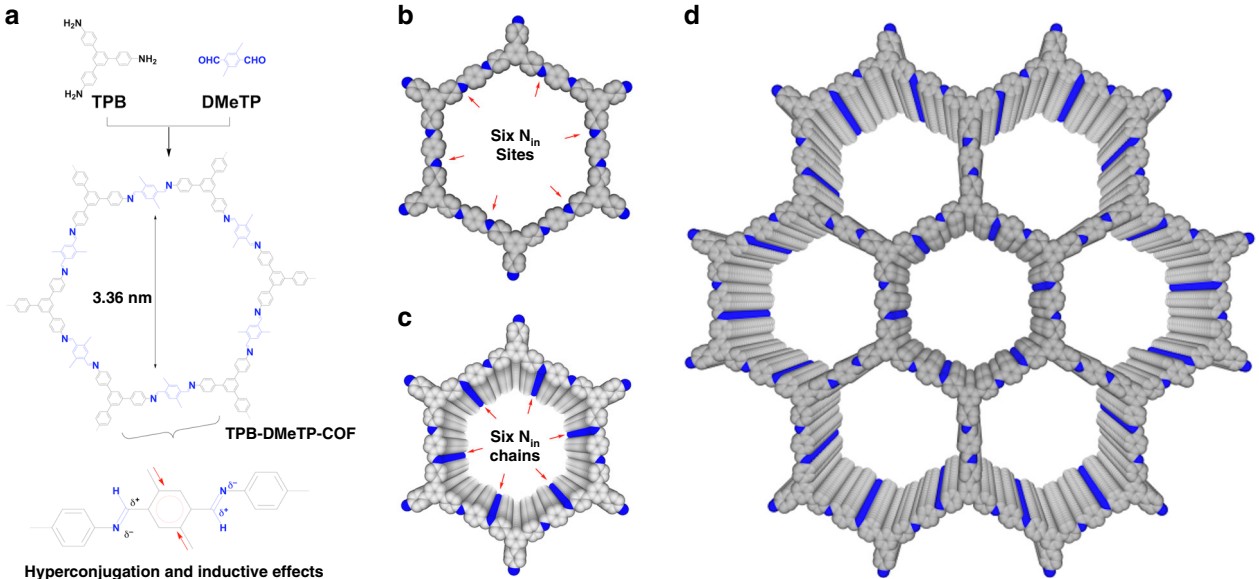

**Fig. 1 Stable covalent organic frameworks with dense 1D nanochannels. a** Topology-guided synthesis of a stable crystalline mesoporous TPB-DMeTP-COF with a pore size of 3.36 nm. Inset shows the induciue and hyperconjugation effects of methyl groups that reduces the polarisation of C=N bonds, softens interlayer charge repulsion and yields an extremely stable framework. **b** Reconstructed structure of one hexagonal macrocycle (grey, C; blue, N; $CH_3$ units and H are omitted for clarity). One hexagon has six $N_{in}$ sites on each edge. **c** Reconstructed structure of a 1D channel; only 20 layers are shown. The 1D channel possesses six $N_{in}$ chains with one on each on pore walls; the $N_{in}$ site is separated by 3.52 Å along the z direction. **d** Reconstructed structure of extended and ordered 1D hexagonal nanochannels; only seven channels and 20 layers are shown. Nanochannels are independent and densely aligned across the material.

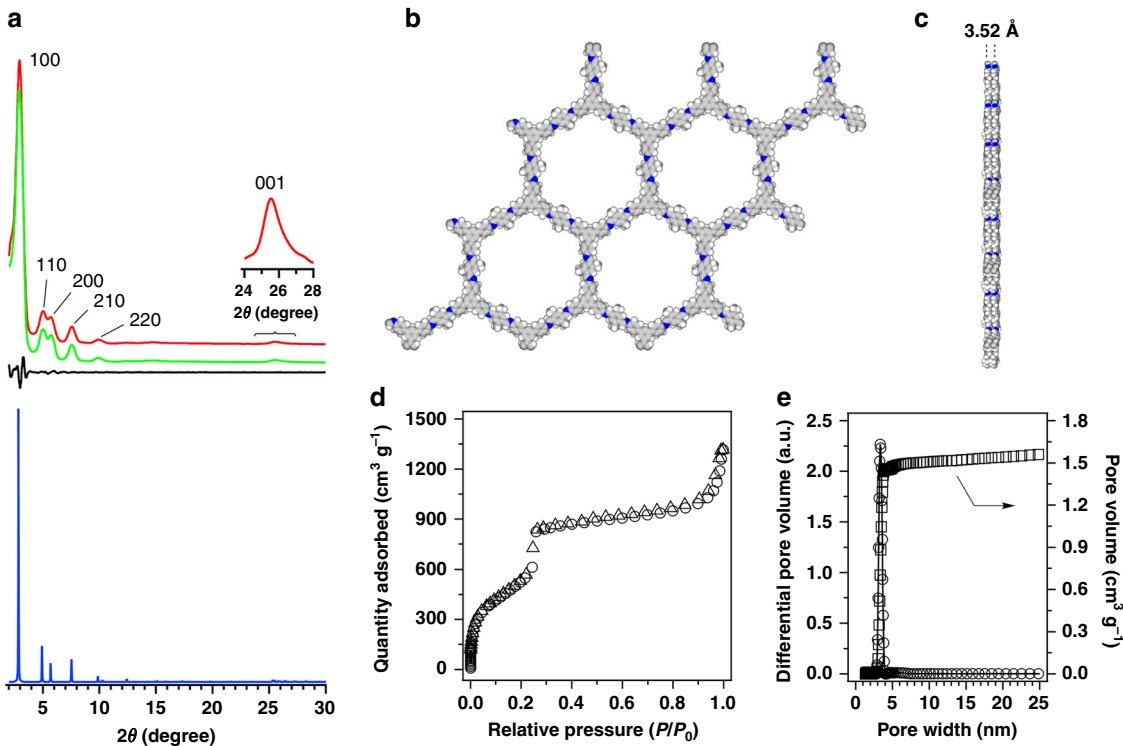

**Fig. 2 Crystallinity and porosity. a** PXRD patterns of TPB-DMeTP-COF (red curve), the Pawley refinement result (green curve) and their difference (black curve), the AA-stacking mode of the $P_6$ space group (blue curve). **b, c** Reconstructed crystal structure of (b) top and (c) side views. The 2D layers are stacked at a 3.52-Å interval along the z direction. **d** Nitrogen sorption isotherms of TPB-DMeTP-COF measured at 77 K (circle, adsorption; triangle, desorption). **e** Pore size distribution (circles) and pore volume (squares) profiles.

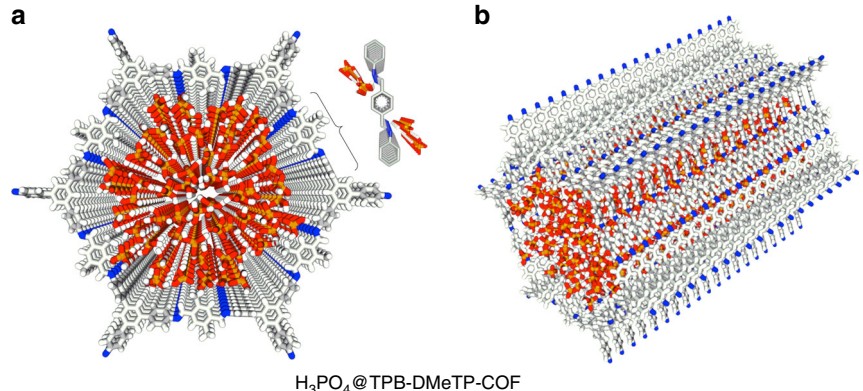

H$_3$PO$_4$@TPB-DMeTP-COF

**Fig. 3 Confining proton network in nanochannel. a** Reconstructed crystal structure of H$_3$PO$_4$@TPB-DMeTP-COF (top view) with the H$_3$PO$_4$ network locked in the 1D channel (each macrocycle of one layer contains 57 H$_3$PO$_4$ molecules; only 20 layers are shown; grey, C; blue, N; white, H; red, O; orange, P). Inset, the N$_{in}$···H–O hydrogen-bonding interactions between channel walls and H$_3$PO$_4$ network. **b** Reconstructed crystal structure of H$_3$PO$_4$@TPB-DMeTP-COF (side view).

We unambiguously characterised the chemical structure of TPB-DMeTP-COF by various analytic methods (Supplementary Figs. 1–12, Supplementary Table 4). Fourier transform Infrared spectroscopy (FT IR) revealed a band at 1618 cm$^{-1}$ that was assigned to the vibration band of the C=N linkage (Supplementary Fig. 1a, red curve). Thermogravimetric analysis indicates that TPB-DMeTP-COF is stable up to 440 °C under nitrogen (Supplementary Fig. 2). Solid-state $^{13}$C pulse magic angle spinning nuclear magnetic resonance spectroscopy (CP/MAS NMR) shows a signal at 157.5 parts per million (ppm), which can be assigned to the carbon atom of C=N unit (Supplementary Fig. 3).

**Stability tests**. We examined the chemical stability by immersing TPB-DMeTP-COF in tetrahydrofuran, H$_3$PO$_4$ (0.7 M in THF), acetonitrile, water (25 and 100 °C) and aqueous HCl (12 M) and NaOH (14 M) solutions for 7 days and observed that TPB-DMeTP-COF is stable to retain its crystallinity (Supplementary Fig. 4a) and porosity (Supplementary Figs. 4b, 4c, 5 and 6, Supplementary Table 4). We further investigated the anti-oxidation stability by Fenton test and confirmed that TPB-DMeTP-COF hardly change its crystallinity and porosity (Supplementary Fig. 7 and Supplementary Table 4).

**Proton transport**. We loaded H$_3$PO$_4$ to the channels of TPB-DMeTP-COF to prepare H$_3$PO$_4$@TPB-DMeTP-COF using neat H$_3$PO$_4$ crystal via a vacuum impregnation method (see Methods). Based on a standard titration protocol (Supplementary Fig. 8)[23], the H$_3$PO$_4$ content in H$_3$PO$_4$@TPB-DMeTP-COF was quantitatively analysed to be 266.6 wt%, which is nearly the same as the maximum loading content (269.6 wt%) according to the pore volume of TPB-DMeTP-COF (1.60 cm$^3$ g$^{-1}$) and the density of H$_3$PO$_4$ (1.685 g cm$^{-3}$). The resulting H$_3$PO$_4$@TPB-DMeTP-COF has almost no crystallinity (Supplementary Fig. 9a, black curve) and is nonporous as the pores are fully occupied by amorphous H$_3$PO$_4$ (Supplementary Fig. 9b). Field-emission scanning electron microscopy (FE SEM) revealed that the morphology of TPB-DMeTP-COF was retained in H$_3$PO$_4$@TPB-DMeTP-COF (Supplementary Fig. 10). X-ray photoelectron spectroscopy (XPS) confirmed the presence of phosphorus atoms (Supplementary Fig. 11). Energy-dispersive X-ray analysis revealed a homogenous distribution of P atoms across the sample (Supplementary Fig. 12).

FT IR spectra revealed the hydrogen-bonding interactions between the channel walls and H$_3$PO$_4$ as evident by a 27-cm$^{-1}$ shift for the vibrational band of C=N bonds from 1618 cm$^{-1}$ of TPB-DMeTP-COF to 1645 cm$^{-1}$ of H$_3$PO$_4$@TPB-DMeTP-COF (Supplementary Fig. 1b)[18]. Molecular dynamics calculations[24] confirmed that each layer can confine 57 H$_3$PO$_4$ molecules to form extended hydrogen-bonding network (Figs. 3a and 3b) with an average binding energy of 70.1 kJ mol$^{-1}$ over the averaged trajectories of H$_3$PO$_4$. Meanwhile, a single point N$_{in}$···H–O hydrogen-bonding interaction between the N$_{in}$ site and H$_3$PO$_4$ yields a binding energy as high as 49.6 kJ mol$^{-1}$, the 3D multichain multipoint interactions across the whole channel are strong enough to confine and stabilise the H$_3$PO$_4$ network.

The TPB-DMeTP-COF itself is an insulator (conductivity = 9.6 × 10$^{-11}$ S cm$^{-1}$ at 160 °C). In impedance spectra (Fig. 4a–h), H$_3$PO$_4$@TPB-DMeTP-COF at 160 °C yields a curve with an intersection at the x axis to give a resistance of 1.03 Ω, from which the anhydrous proton conductivity (σ) is calculated to be as high as 1.91 × 10$^{-1}$ S cm$^{-1}$ (Fig. 4g). The high performance is stable upon continuous run over 20 h (Fig. 4h). H$_3$PO$_4$@TPB-DMeTP-COF enables a proton flow that is even twofold compared to molten neat H$_3$PO$_4$ (~1 × 10$^{-1}$ S cm$^{-1}$)[25]. Note that the proton flow is significantly increased by 2–8 orders of magnitude higher than those of other analogues under anhydrous condition (Table 1)[3,5,6,10,21,26–45]. Meta-organic frameworks (MOFs) are hardly efficient to construct anhydrous proton-conducting systems by loading neat H$_3$PO$_4$ into their pores. Proton transport in 1D channel of COFs is different from that in thermoset polymers such as polybenzimidazoles[10,39,46]; the 1D channel of COFs contains a pure H$_3$PO$_4$ network that is zipped up on the pore walls while the thermoset polymers will be swollen to enhance both the steric hindrance and the system viscosity that lead to block proton motion; this is the reason why these polymers always have one order of magnitude low conductivity compared to neat H$_3$PO$_4$.

The proton conductivity is dependent on temperature and is 1.60 × 10$^{-1}$, 1.34 × 10$^{-1}$, 1.03 × 10$^{-1}$, 7.59 × 10$^{-2}$, 5.90 × 10$^{-2}$ and 4.43 × 10$^{-2}$ S cm$^{-1}$ at 150, 140, 130, 120, 110 and 100 °C, respectively (Table 2). These data confirmed that H$_3$PO$_4$@TPB-DMeTP-COF with dense and large pores enables super proton flow over a wide range of temperature. Note that a recent example of NKCOF-1 with H$_3$PO$_4$ shows no proton conductivity under anhydrous conditions[27]. TPB-DMeTP-COF retains its crystallinity and porosity after AC impedance for 20 h (Supplementary Fig. 13 and Supplementary Table 4).

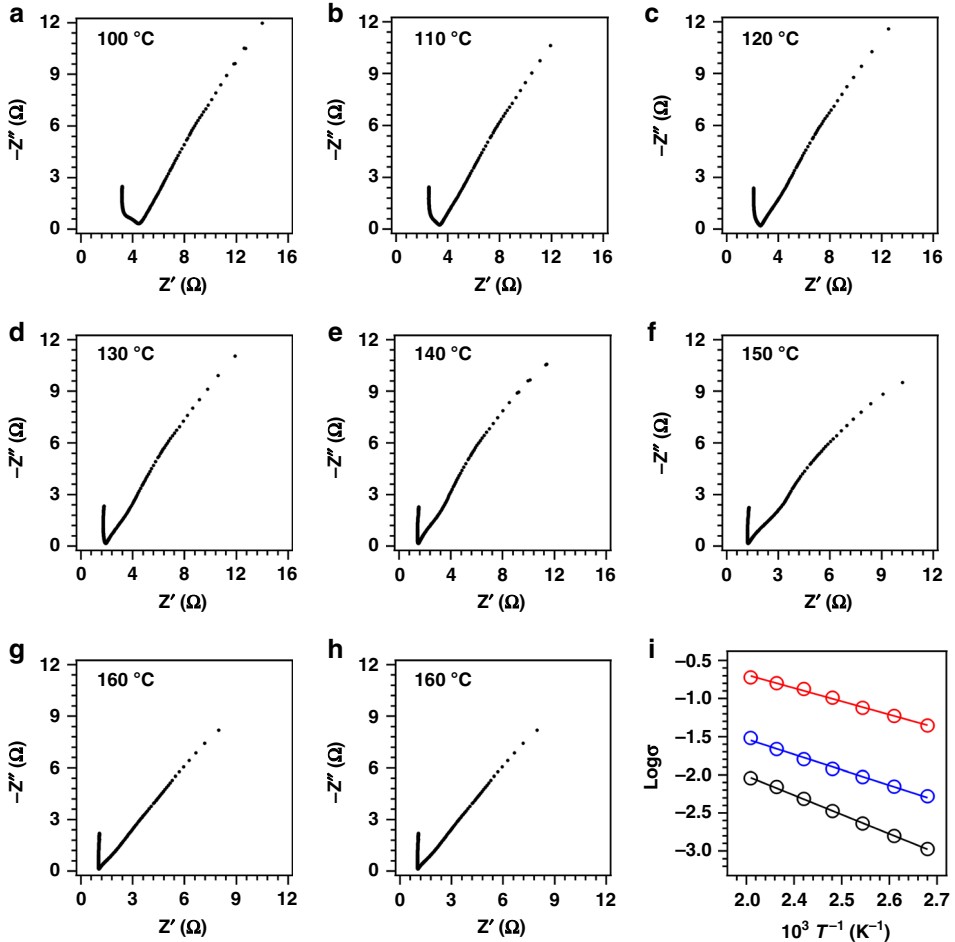

**Fig. 4 Proton transport and activation energy. a–g**, Nyquist plots of $H_3PO_4$@TPB-DMeTP-COF measured at different temperatures under anhydrous conditions. **h** Nyquist plot of $H_3PO_4$@TPB-DMeTP-COF after 20-h continuous run at 160 °C. **i** Temperature profiles of proton conductivities of $H_3PO_4$@TPB-DMeTP-COF (red circles and line), 75%$H_3PO_4$@TPB-DMeTP-COF (blue circles and line) and 50%$H_3PO_4$@TPB-DMeTP-COF (black circles and line). Circles are experimental data and lines are curve fitting.

Proton conduction is a thermal activation process and the conductivity ($\sigma$) can be described by the Arrhenius equation of $\sigma(T) = \sigma_0 \, e^{-E_a/RT}$, where $\sigma_0$ is the pre-exponential factor, $E_a$ is the activation energy (eV), $R$ is the universal gas constant (= 8.3144 J mol K$^{-1}$) and $T$ is the absolute temperature (K). Plotting $\log \sigma$ versus temperature ($T^{-1}$) yields a linear curve (Fig. 4i, red circles and curve); from the slope, the activation energy ($E_a$) is evaluated to be 0.34 eV. This indicates a low-energy hopping over the proton network confined in the nanochannels[47]. Molecular dynamics at sub picosecond precision revealed that the proton diffusion rate is $6.28 \times 10^{-4}$ cm$^2$ s$^{-1}$ at 383 K (Supplementary Table 5), which is much higher than that ($4.0 \times 10^{-6}$ cm$^2$ s$^{-1}$)[20] of molten neat $H_3PO_4$. Moreover, the reorganisation energy of $H_3PO_4$ in proton flow is only 0.12 eV. The high mobility together with an extremely low reorganisation energy enables super flow.

## Discussion

We further synthesised two COF samples with different $H_3PO_4$ contents to disclose the impact of the $H_3PO_4$ network on proton flow. The 75%$H_3PO_4$@TPB-DMeTP-COF samples with three-fourth of the theoretical $H_3PO_4$ loading content (Supplementary Figs. 9a (green curve), 10, 11, 14 and 15a) exhibit proton conductivities of $3.05 \times 10^{-2}$, $2.19 \times 10^{-2}$, $1.60 \times 10^{-2}$, $1.20 \times 10^{-2}$,

$9.41 \times 10^{-3}$, $7.01 \times 10^{-3}$ and $5.28 \times 10^{-3}$ S cm$^{-1}$ at 160, 150, 140, 130, 120, 110 and 100 °C, respectively (Table 2 and Supplementary Fig. 17). Temperature-dependent proton conductivities revealed that the $E_a$ value is 0.40 eV (Fig. 4i, blue circles and curve). The 50%$H_3PO_4$@TPB-DMeTP-COF samples with half of the theoretical $H_3PO_4$ loading content (Supplementary Figs. 9a (blue), 10, 11, 15b and 16) exhibit proton conductivities of $9.04 \times 10^{-3}$, $6.94 \times 10^{-3}$, $4.88 \times 10^{-3}$, $3.36 \times 10^{-3}$, $2.30 \times 10^{-3}$, $1.59 \times 10^{-3}$ and $1.07 \times 10^{-3}$ S cm$^{-1}$ at 160, 150, 140, 130, 120, 110 and 100 °C, respectively (Table 2 and Supplementary Fig. 18), while the $E_a$ value is increased to 0.50 eV (Fig. 4i, black circles and curve). The decreasing tendency of proton transport with loading content is much small compared to those of engineering plastics such as polybenzimidazoles;[10,39,46] this originates from the fact that the $N_{in}$ sites on the channel walls are dense along the $z$ direction so that a continuous $H_3PO_4$ network can still form proximate to the $N_{in}$ chains even at a low loading content, which offers the pathways for proton flow; however, this feature is unavailable for polybenzimidazoles (Fig. 3a, inset).

We have demonstrated a strategy for designing proton-conducting materials based on stable COFs by exploring their well-defined 1D nanochannels to preset hydrogen-bonding sites on the pore walls so that both COF and proton network can be stabilised to achieve exceptional stability with super proton flow in one material. Through the mechanistic and comparative

**Table 1 Comparison of proton conductivities in some reported materials using phosphoric acid or organic heterocyclic derivatives as proton carriers.**

| Material | Proton conductivities (S cm$^{-1}$) | Temperature (°C) | Ref. |
|---|---|---|---|
| *Pure H$_3$PO$_4$* | | | |
| H$_3$PO$_4$@TPB-DMeTP-COF | $1.91 \times 10^{-1}$ | 160 | This work |
| H$_3$PO$_4$@Tp-Azo-COF | $6.70 \times 10^{-5}$ | 67 | Ref. [6] |
| H$_3$PO$_4$@TpBpy-MC | $2.50 \times 10^{-3}$ | 120 | Ref. [28] |
| H$_3$PO$_4$/(PEO-PMA)/PEGDE | $1.30 \times 10^{-4}$ | 90 | Ref. [29] |
| H$_3$PO$_4$/meso-silica | $6.00 \times 10^{-2}$ | 225 | Ref. [3] |
| H$_3$PO$_4$/PBI-ZIF(8/67) | $9.20 \times 10^{-2}$ | 200 | Ref. [30] |
| H$_3$PO$_4$ doped Porous PBI | $5.00 \times 10^{-2}$ | 140 | Ref. [31] |
| H$_3$PO$_4$ doped polyimide | $1.00 \times 10^{-4}$ | 140 | Ref. [32] |
| H$_3$PO$_4$ doped PAM | $6.35 \times 10^{-2}$ | 183 | Ref. [33] |
| H$_3$PO$_4$@NKCOFs | negligible | 80 | Ref. [27] |
| Neat H$_3$PO$_4$ | ~$1.0 \times 10^{-1}$ | 160 | Ref. [25] |
| aza-COF-2$_H$ | $4.80 \times 10^{-3}$ | 50 (97% RH) | Ref. [5] |
| *Phytic acid* | | | |
| Phytic acid@TpPa-(SO$_3$H-Py) | $3.00 \times 10^{-4}$ | 120 | Ref. [40] |
| *PTSA·H$_2$O* | | | |
| PTSA@TpAzo | $7.8 \times 10^{-2}$ | 80 (95% RH) | Ref. [41] |
| *H$_2$O* | | | |
| NUS-10(R) | $3.96 \times 10^{-2}$ | 25 (97% RH) | Ref. [42] |
| LiCl@RT-COF-1 | $6.45 \times 10^{-3}$ | 40 (100% RH) | Ref. [43] |
| Benzimidazole-linked 2D-polymers | $3.2 \times 10^{-2}$ | 95 (95% RH) | Ref. [44] |
| EB-COF:PW$_{12}$ | $3.32 \times 10^{-6}$ | 20 (97% RH) | Ref. [45] |
| *Organic heterocycles* | | | |
| Im@TPB-DMTP-COF | $4.37 \times 10^{-3}$ | 130 | Ref. [4] |
| Tri@TPB-DMTP-COF | $1.10 \times 10^{-3}$ | 130 | Ref. [4] |
| HL@0.202Him | $6.57 \times 10^{-5}$ | 120 | Ref. [34] |
| [Al(OH)(ndc)]n⊃im | $2.20 \times 10^{-5}$ | 120 | Ref. [35] |
| Im@Td-PPI | $3.49 \times 10^{-4}$ | 90 | Ref. [36] |
| Im@Td-PNDI | $9.04 \times 10^{-5}$ | 90 | Ref. [36] |
| [Al(OH)(ndc)]n⊃His | $1.70 \times 10^{-3}$ | 150 | Ref. [37] |
| [Zn$_3$(HPO$_4$)$_6$(H$_2$O)$_3$](Hbim) | $1.30 \times 10^{-3}$ | 120 | Ref. [38] |

studies, it becomes clear that the criteria of a porous material for anhydrous proton transport is threefold: (i) the stability of the framework; (ii) the stability of proton network in the pores; and (iii) the porosity of material that determines the density of proton networks in the material. Our results suggest that proton flow is correlated with the continuum of the H$_3$PO$_4$ network in the nanochannels; a full loading fabricates a continuous proton network and enables super flow across the material. This scenario for proton transport is distinct from a single submicropore in biological ion channels and demonstrates that large mesopores (>20 Å) also enable proton super flow by working on a different molecular mechanism. Moreover, from the perspective of chemistry, the 1D nanochannels based on COFs are more chemically designable and synthetically accessible. We anticipate that this study would be a start line for anhydrous proton-conducting materials and their implementations in energy conversion and storage.

## Methods

**Chemicals**. 1,4-Dimethylbenzene and 1,3,5-tri(4-aminophenyl) benzene (TAPB) were purchased from TCI and used as received. Crystalline neat phosphoric acid was purchased from Sigma-Aldrich and used as received. Tetrahydrofuran (THF) was refluxed with sodium and benzophenone to remove water and oxygen before use. Dehydrated *N, N*-dimethylformide (DMF), acetonitrile (CH$_3$CN) and *o*-dichlorobenzene (*o*-DCB) were purchased from Kanto Chemicals. Dioxane, mesitylene, *n*-butanol and acetic acid were purchased from Wako Chemicals. 2,5-Dimethylterephthalaldehyde (DMeTP) was synthesised according to the reported method[48].

**Synthesis of TPB-DMeTP-COF**. An *o*-DCB/*n*-BuOH (0.5 mL/0.5 mL) mixture of TAPB (0.094 mmol, 32.7 mg), DMeTP (0.141 mmol, 22.8 mg) and acetic acid (6 M, 0.1 mL) in a Pyrex tube (10 mL) was degassed via three freeze–pump–thaw cycles. The tube was flame sealed and heated at 120 °C for 3 days. The precipitate was collected via centrifugation, washed six times with THF and then subjected to Soxhlet extraction with THF for 1 day to remove any trapped guest molecules. The powder was collected and dried at 120 °C under vacuum overnight to produce TPB-DMeTP-COF in an isolated yield of 89%.

**Synthesis of H$_3$PO$_4$@TPB-DMeTP-COF**. Neat H$_3$PO$_4$ was impregnated into TPB-DMeTP-COF via vacuum assisted method as shown below. Crystalline neat phosphoric acid (269.6 mg) was dissolved in anhydrous THF (4 mL). The resulting homogeneous solution was injected into the TPB-DMeTP-COF sample (100 mg) in a vial (20 mL) preheated under vacuum at 120 °C overnight to yield a solution which was kept to stir at room temperature for 3 h. The system was slowly evaporated under vacuum to remove THF at 70 °C over a period of 6 h under vacuum. The vial was then kept in an oven at 70 °C for 12 h. The resulting powder was collected to yield H$_3$PO$_4$@TPB-DMeTP-COF quantitatively. The 75% H$_3$PO$_4$@TPB-DMeTP-COF and 50%H$_3$PO$_4$@ TPB-DMeTP-COF were synthesised using the same VAM process except the different loading amounts of crystalline neat H$_3$PO$_4$.

**Stability test**. The COF samples (100 mg) were dispersed in different solvents including water (25 and 100 °C), H$_3$PO$_4$ (0.7 M, a THF solution (4 mL) of H$_3$PO$_4$ (270 mg), THF, CH$_3$CN and aqueous HCl (12 M) and NaOH (14 M) solutions and stirred for one week. Before PXRD and nitrogen sorption isotherm measurements, the resulting COF samples were treated as follow. The COF samples in water, THF and CH$_3$CN were collected and dried at 120 °C under vacuum for 12 h. The COF samples in THF solution of H$_3$PO$_4$ and aqueous HCl solution were washed with a large amount of water, neutralised with triethylamine, rinsed with water and ethanol, and dried under vacuum at 120 °C for 12 h. The COF sample in the aqueous NaOH solution was washed with a large amount of water and THF and dried under vacuum at 120 °C for 12 h.

**Fenton test**. The COF sample (50 mg) was kept in Fenton's reagent[49] (20 mL, 3% H$_2$O$_2$, 3 ppm Fe(II)) for 24 h. The resulting sample was washed with THF, dried under vacuum at 120 °C for 12 h and subjected to PXRD and nitrogen sorption isotherm measurements.

**Method for analysis the content of H$_3$PO$_4$ in COFs**. The exact content of H$_3$PO$_4$ in H$_3$PO$_4$@TPB-DMeTP-COF was quantitatively analysed using the standard method[23] via the following reaction:

$$PO_4^{3-} + 12 \ (NH_4)_2MoO_4 + 21 \ NO_3^- + 24 \ H^+ = (NH_4)_3PO_4 \cdot 12MoO_3 \downarrow + 21 \ NH_4NO_3 + 12 \ H_2O$$

A H$_3$PO$_4$@TPB-DMeTP-COF sample (155 mg) was added to an aqueous NaOH solution (130 mg, 0.23 mM, 14 mL) and the mixture was stirred for 10 h. The mixture was filtrate and the solid was washed with deionised water till pH = 7. The solution was collected and added with HNO$_3$ till pH = 2, to which an aqueous solution of (NH$_4$)$_2$MoO$_4$ (2276.2 mg, 15 mL) was slowly added to yield a canary yellow precipitate (NH$_4$)$_3$PO$_4$·12MoO$_3$. The resulted (NH$_4$)$_3$PO$_4$·12MoO$_3$ solid was collected by filtration, washed with deionised water (200 mL) and dried under vacuum at 120 °C overnight to yield a total amount of 1795.8 mg. Using this method, the H$_3$PO$_4$ contents of H$_3$PO$_4$@TPB-DMeTP-COF, 75%H$_3$PO$_4$@TPB-DMeTP-COF and 50%H$_3$PO$_4$@TPB-DMeTP-COF were determined to be 266.6, 200.2 and 133.5 wt%, which corresponds to 100%, 75 and 50% occupation of the pore volume of TPB-DMeTP-COF by H$_3$PO$_4$, respectively.

**Structure and characterisations**. Solid-state $^{13}$C cross-polarisation magic angle spinning nuclear magnetic resonance ($^{13}$C CP/MAS NMR) spectra were recorded on a Bruker model biospin AvanceIII500 (500 MHz) NMR spectrometer using the rotor frequency of 10 kHz. Powder X-ray diffraction (PXRD) data were recorded on a Rigaku model Smart Lab X-ray diffractometer ($\lambda = 1.5418$ Å at 30 kV and 30 mA) by depositing samples on glass substrate, scanning from $2\theta = 1.0°$ up to 30° with 0.02° increment. Nitrogen sorption isotherms were measured at 77 K with a Micromeritics Instrument Corporation model 3 Flex surface characterisation analyser. The samples were degassed at 120 °C for 12 h before the measurements. N$_2$ isotherms were recorded at 77 K by using ultra-high purity N$_2$ (99.999% purity). The BET and Langmuir methods were utilised to calculate the specific surface areas and the pore volume were determined using the DFT pore size model. Fourier transform infrared (FT IR) spectra were recorded on a JASCO model FT-IR-6100

**Table 2 Anhydrous proton conductivity of H$_3$PO$_4$@TPB-DMeTP-COF, 75%H$_3$PO$_4$@TPB-DMeTP-COF and 50%H$_3$PO$_4$@TPB-DMeTP-COF at different temperatures.**

| COF samples | Proton conductivities (S cm$^{-1}$) | | | | | | |
|---|---|---|---|---|---|---|---|
| | 100 °C | 110 °C | 120 °C | 130 °C | 140 °C | 150 °C | 160 °C |
| H$_3$PO$_4$@TPB-DMeTP-COF | 0.0443 | 0.0590 | 0.0759 | 0.103 | 0.134 | 0.160 | 0.191 |
| 75%H$_3$PO$_4$@TPB-DMeTP-COF | 0.00528 | 0.00701 | 0.00941 | 0.0120 | 0.0160 | 0.0219 | 0.0305 |
| 50%H$_3$PO$_4$@TPB-DMeTP-COF | 0.00107 | 0.00159 | 0.00230 | 0.00336 | 0.00488 | 0.00694 | 0.00904 |

infrared spectrometer using KBr pellets. Field emission scanning electron microscopy (FE SEM) images were obtained on a Hitachi high technologies model field-emission scanning electron microscope SU-6600 at a voltage of 5 kV. X-ray photoelectron spectroscopy (XPS) experiments were carried out on an AXIS Ultra DLD system from Kratos with Al Kα radiation as X-ray source. Thermogravimetric analysis (TGA) measurements were performed on a Mettler-Toledo model TGA/SDTA851e under N$_2$, by heating to 800 °C at a rate of 10 °C min$^{-1}$. Energy dispersive X-ray analysis (EDX) and elemental mapping were acquired on a Hitachi model Miniscope TM3030.

**Crystal structural reconstruction**. The crystalline structure of TPB-DMeTP-COF was determined by using the PXRD pattern in conjunction with the density-functional tight-binding (DFTB+) method including Lennard-Jones (LJ) dispersion. The calculations[22] were carried out with the DFTB+ program package version 1.2. DFTB is an approximate density functional theory method based on the tight-binding approach and utilises an optimised minimal LCAO Slater-type all-valence basis set in combination with a two-centre approximation for Hamiltonian matrix elements. The Columbic interaction between partial atomic charges was determined using the self-consistent charge (SCC) formalism. Lennard-Jones type dispersion was employed in all calculations to describe Van der Waals (VDW) and π-stacking interactions. The lattice dimensions were optimised simultaneously with the geometry. Standard DFTB parameters for X–Y element pair (X, Y=C, H and N) interactions were employed from the mio-0-1 set[50].

Molecular modelling and Rietveld refinement were carried out using Reflex, a software package for crystal determination from PXRD pattern, implemented in MS modelling version 8.0 (Accelrys Inc.). Unit cell dimension was first manually determined from the observed PXRD peak positions using the coordinates. We performed Pawley refinement to optimize the lattice parameters iteratively until the $R_{wp}$ value converges. The pseudo-Voigt profile function was used for whole profile fitting and Berrar–Baldinozzi function was used for asymmetry correction during the refinement processes. The final $R_{wp}$ and $R_p$ values were 2.01% and 1.12%, respectively.

**Molecular dynamics calculations**. From the lowest energy AA geometry of TPB-DMeTP-COF, ten sets of initial coordinates of H$_3$PO$_4$@TPB-DMeTP-COF were generated and their geometries optimised, keeping the lattice parameters of the COF fixed. The three lowest energy configurations were then used for molecular dynamics calculations. Firstly, the coordinates were pre-equilibrated in the canonical (NVT) ensemble for 10 ps. Each nuclear degree of freedom was coupled to a Nose-Hoover chain thermostat with the target kinetic temperature set to 383 K. After 10 ps, the thermostats were turned off and the system was allowed to evolve in the microcanonical (NVE) ensemble for 100 ps at an average kinetic temperature of 383 K. All calculations[22] used the Self-Consistent-Charge Density-Functional Tight-Binding (SCC-DFTB) method with Lennard-Jones dispersion, as implemented in DFTB+ 18.1. All atoms (C, N, O, P and H) were described using the mio-1-1 parameter set[51].

To calculate the proton diffusion, the mean square displacements of protons, excluding those of the COF were calculated using ISAACS[52]. The proton diffusion rate was subsequently calculated from the straight-line portion of the MSD, using NumPy polyfit package. The hydrogen-bonding network was generated for each step of the molecular dynamics calculation, using the neighbour list function provided in the python Atomic Simulation Environment (ASE) version 3.17[53]. The script is available at https://github.com/maddicoat/h-bonds. Briefly, the initial state was assumed to correspond to 57 distinct H$_3$PO$_4$ molecules as initially generated, corresponding to the real density of H$_3$PO$_4$ integrated into the channel. A neighbour list was generated for each set of saved coordinates and any hydrogen bonds (≤1.8 Å) between H atoms of one H$_3$PO$_4$ molecule and O atoms of another were noted in a H$_3$PO$_4$ connectivity matrix, which was then used to derive the hydrogen-bonding network. The H$_3$PO$_4$ reorganisation energy (0.118 eV) was calculated in AMS 2019 using the BLYP-D3 functional with a TZP basis set[24].

**Impedance spectroscopy**. The H$_3$PO$_4$@TPB-DMeTP-COF samples were pressed into thin pellets as follow. The samples were well ground into homogeneous powder in a mortar by a pestle under dry nitrogen. The powder was mounted quickly to the mould with a standard die of 10-mm diameter, sandwiched

between two steel mirrors, and pressed using an oil press equipment Riken Power model P-1B-041B 200KN at 100 kN for 30 min under dry nitrogen. Impedance measurements were conducted on a HIOKI model IM3570 impedance analyser, with a two-probe electrochemical cell with the frequency range 4 Hz to 500 MHz and an input voltage 100 mV. The measurements were performed under dry nitrogen.

## Data availability

All data needed to evaluate the conclusions given in the paper are present in the paper and Supplementary Information. Any additional data related to this paper may be requested from the authors.

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

## Acknowledgements

D.J. acknowledges supports by MOE tier 1 grant (R-143-000-A71-114) and NUS start-up grant (R-143-000-A28-133). M.A.A. acknowledges support from EPSRC, EP/S015868/1 and HPC resources on THOMAS via the Materials Chemistry Consortium, EP/P020194.

## Author contributions

D.J. conceived and designed the project. S.T., L. Z. and Q.J. conducted the experiments. A.D.D.W. and M.A.A. carried out DFTB calculations and molecular dynamics evaluations. D.J. and S.T. wrote the manuscript and discussed the results with the contributing authors. All data are reported in the main text and supplement.

## Competing interests

The authors declare no competing interests.
