## [Peer Review File · Nature Communications]

Reviewers' comments:

Reviewer #1 (Remarks to the Author):

The manuscript presents a novel strategy to fabricate proton conductive materials based on robust COFs by exploring their well-defined 1D nanochannels to preset hydrogen-bonding sites on the pore walls. Thus, both COF and proton network can be stabilized to achieve exceptional stability with super proton flow in one material. To the best of our knowledge, the achieved proton conductivity is the highest in the COF field. The author also gave an in-depth explanation for the outstanding proton conductivity. Overall, the scientific level of this manuscript is very high and can fit the criteria of the journal. I highly recommend it to be accepted after minor revision. Some considerations before final acceptance :

1. Please provide the PXRD data of H₃PO₄@TPB-DMeTP-COF.
2. After the AC impedance measurement, please verify the stability of TPB-DMeTP-COF
3. Please explain the role of meso-sized pore of TPB-DMeTP-COF playing in the proton conduction.
4. Please describe the concentration of H₃PO₄ in the stability evaluation part.

Reviewer #2 (Remarks to the Author):

This is an excellent work carried out in a field of high scientific interest with potential technological impact. The experimental data corroborate the claims. The results are outstanding showing a designed COF material with high proton conductivity, proton super flow, even under dried conditions. I can highly recommend publication after minor corrections.

- 1) In the Section called "Proton Transport" de authors mention "vacuum impregnation method": I cannot find a suitable description of this method.
- 2) The sentence "Molecular dynamics calculations confirmed that each layer can confine 57 H₃PO₄ molecules to form extended hydrogen-bonding network" looks rather unclear to me. Can the authors provide the number of H₃PO₄ molecules per repetition unit? 57 molecules per layer is not clear if they do not provide more information.
- 3) Have the authors perform electrical measurements at variable humidity?
- 4) In the literature I miss some references of ionic conductivity in COFs for data comparison (this could be added in table 2).

Reply to Reviewer #1

The manuscript presents a novel strategy to fabricate proton conductive materials based on robust COFs by exploring their well-defined 1D nanochannels to preset hydrogen-bonding sites on the pore walls. Thus, both COF and proton network can be stabilized to achieve exceptional stability with super proton flow in one material. To the best of our knowledge, the achieved proton conductivity is the highest in the COF field. The author also gave an in-depth explanation for the outstanding proton conductivity. Overall, the scientific level of this manuscript is very high and can fit the criteria of the journal. I highly recommend it to be accepted after minor revision.

We appreciate these comments.

Some considerations before final acceptance:

1. Please provide the PXRD data of H₃PO₄@TPB-DMeTP-COF.

We appreciate this comment.

We have added the PXRD patterns of H₃PO₄@TPB-DMeTP-COF, 75%H₃PO₄@TPB-DMeTP-COF and 50%H₃PO₄@TPB-DMeTP-COF to the revised Supplementary Information Fig. 9a.

We have added a sentence “The resulting H₃PO₄@TPB-DMeTP-COF has almost no crystallinity (Supplementary Fig. 9a, black curve) and becomes nonporous as the pores are fully occupied by amorphous H₃PO₄ (Supplementary Fig. 9b).”, to the revised manuscript.

2. After the AC impedance measurement, please verify the stability of TPB-DMeTP-COF.

We appreciate this comment.

We have confirmed the stability of TPB-DMeTP-COF after the AC impedance. The H₃PO₄ was removed by washing with large amount of water, then neutralised with NaOH, rinsed with water and ethanol, and then Soxhlet with THF for 24 h. The powder was collected and dried under vacuum at 120 °C for 12 h before measuring the crystallinity and porosity. The results revealed that the crystallinity and porosity are retained after the AC impedance; the BET surface area was 2292 m² g⁻¹ with the pore size of 3.33 nm and pore volume of 1.22 cm³ g⁻¹. We have added Figure S13 and porosity data (Table S4) to the revised Supplementary Information.

We have added a sentence “TPB-DMeTP-COF retains its crystallinity and porosity after AC impedance for 20 h (Supplementary Fig. 13 and Supplementary Table 4).”, to the revised manuscript.

3. Please explain the role of meso-sized pore of TPB-DMeTP-COF playing in the proton conduction.

We appreciate this comment.

The functions of meso-sized pore of TPB-DMeTP-COF are multi-fold. (1) It offers the space for loading H₃PO₄; (2) it provides pore wall interface to confine and lock the H₃PO₄ networks; (3) it forms the pathway for proton transport. Without these multi-fold functionalities in the channel, the COF cannot achieve such a high proton conductivity.

4. Please describe the concentration of H₃PO₄ in the stability evaluation part.

We appreciate this comment.

The concentration of H_3PO_4 in the stability evaluation part is 0.7 M. We have added this information to the revised manuscript.

Reply to Reviewer #2

This is an excellent work carried out in a field of high scientific interest with potential technological impact. The experimental data corroborate the claims. The results are outstanding showing a designed COF material with high proton conductivity, proton super flow, even under dried conditions. I can highly recommend publication after minor corrections.

We appreciate this comment.

1) In the Section called “Proton Transport” de authors mention “vacuum impregnation method”: I cannot find a suitable description of this method.

We appreciate this comment.

In the original manuscript, **Methods** section, we have a paragraph on **Synthesis of H₃PO₄@TPB-DMeTP-COF** in which the vacuum impregnation method was described.

2) The sentence “Molecular dynamics calculations confirmed that each layer can confine 57 H₃PO₄ molecules to form extended hydrogen-bonding network” looks rather unclear to me. Can the authors provide the number of H₃PO₄ molecules per repetition unit? 57 molecules per layer is not clear if they do not provide more information.

We appreciate this comment.

The hexagonal macrocycle has a pore size of 3.33 nm and a thickness of 3.52 Å (layer separation). We added H₃PO₄ molecules to one hexagonal pore of one-layer thickness and the number we can add is 57.

To clarify this sentence, we have revised the sentence with a new one “Molecular dynamics calculations confirmed that each **hexagonal macrocycle of one** layer can confine 57 H₃PO₄ molecules to form extended hydrogen-bonding network.”, by adding the new phrase in red.

We have updated the Figure 3 with a high-resolution image.

3) Have the authors perform electrical measurements at variable humidity?

As we are targeting for anhydrous hydron conduction above 100 °C, this manuscript focuses on the chemistry on anhydrous conduction. Anhydrous proton conduction is much challenging compared to proton conduction under humidity.

Nevertheless, as for proton conduction under various humidity, we hope to report the results in a future manuscript as their chemistry is different from anhydrous conduction.

4) In the literature I miss some references of ionic conductivity in COFs for data comparison (this could be added in table 2).

We have added all references (*Chem. Mater.* 2016, 28,1489-1494, *ACS Appl. Mater. Interfaces* 2016, 8, 28, 18505-18512.; *Chem. Mater.* 2019, 31, 819-825.; *J. Am. Chem. Soc.* 2016 ,138, 5897-5903.; *J. Am. Chem. Soc.* 2017, 139,10079-10086.; *Angew Chem Int Ed.* 2018 ,57,10894-10898.; *J. Am. Chem. Soc.* 2019, 141, 38, 14950-14954.) that were reported to date for ion conduction in COFs to the revised Table 2. Note that these systems are different from that of our system using H₃PO₄ under anhydrous conditions.

REVIEWERS' COMMENTS:

Reviewer #1 (Remarks to the Author):

The authors has correctly answered all questions. Publish as it is.

Reviewer #2 (Remarks to the Author):

The revised version of the manuscript is suitable for publication in its present form.

Point-to-point answers

Reviewer #1 (Remarks to the Author):

The authors has correctly answered all questions. Publish as it is.

Answer: Thank you very much fir your comment.

Reviewer #2 (Remarks to the Author):

The revised version of the manuscript is suitable for publication in its present form.

Answer: Thank you very much fir your comment.